# Neuropeptides’ Hypothalamic Regulation of Sleep Control in Children Affected by Functional Non-Retentive Fecal Incontinence

**DOI:** 10.3390/brainsci10030129

**Published:** 2020-02-25

**Authors:** Vincenzo Monda, Marco Carotenuto, Francesco Precenzano, Diego Iacono, Antonietta Messina, Monica Salerno, Francesco Sessa, Valentina Lanzara, Giovanni Messina, Giuseppe Quatrosi, Rosaria Nardello, Francesca Felicia Operto, Claudia Santoro, Fiorenzo Moscatelli, Chiara Porro, Christian Zammit, Marcellino Monda, Grazia Maria Giovanna Pastorino, Luigi Vetri, Lucia Parisi, Maria Ruberto, Michele Roccella

**Affiliations:** 1Department of Experimental Medicine, Section of Human Physiology and Unit of Dietetic and Sport Medicine, University of Campania “Luigi Vanvitelli”, 81100 Caserta CE, Italy; mondavincenzo@gmail.com (V.M.); Antonietta.messina@unicampania.it (A.M.); marcellino.monda@unicampania.it (M.M.); maria.ruberto@unicampania.it (M.R.); 2Sleep Lab for Developmental Age, Clinic of Child and Adolescent Neuropsychiatry, Department of Mental Health, Physical and Preventive Medicine, University of Campania “Luigi Vanvitelli”, 81100 Caserta CE, Italy; marco.carotenuto@unicampania.it (M.C.); f.precenzano@hotmail.it (F.P.); valelanz87@hotmail.it (V.L.); dr.claudiasantoro@gmail.com (C.S.); graziapastorino@gmail.com (G.M.G.P.); 3Neurodevelopmental Research Lab, Biomedical Research Institute of New Jersey (BRInj), Morristown, NJ 07960, USA; diego.iacono@atlantichealth.org; 4Neuroscience Research, MidAtlantic Neonatology Associates (MANA), Atlantic Health System (AHS), Morristown, NJ 07927, USA; 5Neuropathology Research, MANA-BRInj, Cedar Knolls, NJ 07927, USA; 6Department of Clinical and Experimental Medicine, University of Foggia, 71122 Foggia FG, Italy; monica.salerno@unict.it (M.S.); francesco.sessa@unifg.it (F.S.); fiorenzo400@gmail.com (F.M.); chiara.porro@unifg.it (C.P.); 7Department of Psychology, Educational and Science and Human Movement, University of Palermo, 90133 Palermo PA, Italy; peppe.quatrosi@gmail.com (G.Q.); lucia.parisi@unipa.it (L.P.); michele.roccella@unipa.it (M.R.); 8Department of Sciences for Health Promotion and Mother and Child Care “G. D’Alessandro,” University of Palermo, 90133 Palermo PA, Italy; rosaria.nardello@unipa.it (R.N.); luigi.vetri@gmail.com (L.V.); 9Child and Adolescent Neuropsychiatry Unit, Department of Medicine, Surgery and Dentistry, University of Salerno, 84084 Fisciano SA, Italy; opertofrancesca@gmail.com; 10Department of Woman, Child and General and Specialistc Surgery, University of Campania, “Luigi Vanvitelli”, 81100 Caserta CE, Italy; 11Anatomy Department, Faculty of Medicine and Surgery, University of Malta, MSD 2080 Msida, Malta; christianzammit84@gmail.com

**Keywords:** functional non-retentive fecal incontinence (FNRFI), polysomnographic (PSG) assessment, orexin-A, sleep organization disorders

## Abstract

Functional non-retentive fecal incontinence (FNRFI) is a common problem in pediatric age. FNRFI is defined as unintended loss of stool in a 4-year-old or older child after organic causes have been excluded. FNRFI tends to affects up to 3% of children older than 4 years, with males being affected more frequently than females. Clinically, children affected by FNRFI have normal intestinal movements and stool consistency. Literature data show that children with fecal incontinence have increased levels of separation anxiety, specific phobias, general anxiety, attention-deficit/hyperactivity disorder (ADHD), and oppositional defiant disorder. In terms of possible relationship between incontinence and sleep, disorders of sleep organization have been observed in the pathogenesis of enuresis so generating the hypothesis that the orexinergic system may have a crucial role not only for the sleep organization *per se* but also for the sphincterial control in general. This study aimed to focus on specific neurophysiological aspects to investigate on the possible relationship between sleep organizational abnormalities and FNRFI. Specifically, we aimed to measure orexin serum levels in children with FNRFI and assess their polysomnographic sleep macrostructure patterns. Two study groups were considered: FNFRI (*n* = 45) and typically developed (TD) (*n* = 45) group. In both groups, sleep patterns and respiratory events were assessed by polysomnographic recordings (PSG) during a period of two nights at least, and plasma levels of Orexin-A were measured in each participant. The findings of this initial investigation seem to support a major role of Orexin-A in sleep organization alterations in children with FNFRI. Also, our data suggest that sleep habits evaluation should be considered as screening and complementary tool for the diagnosis of fecal incontinence in children.

## 1. Introduction

Functional non-retentive fecal incontinence (FNRFI) is a relatively common problem in pediatric age [1]. FNRFI is associated with high levels of distress for both children and parents and with emotional disorders in about 30–50% of affected children [2]. Two major forms of fecal incontinence can be differentiated: fecal incontinence with and without constipation. According to the Rome IV criteria [3] fecal incontinence is defined as unintended loss of solid or liquid stool occurring in inappropriate places in a 4-year-old or older child after organic causes have been ruled out.

Although epidemiological data are still contrasting and the estimated prevalence depending on the definition used, FNRFI tends to affects up to 3% of children older than 4 years [4,5]. Moreover, males seem to be affected more frequently than females and daytime FNRFI is more frequent than nocturnal FNRFI, which is most often due to organic causes [6].

Children affected by FNRFI have normal intestinal movements and stool consistency, infrequent abdominal pain, normal colon transit and no stool mass. In general, coordination of pelvic visceral activities allowing appropriate elimination behaviors requires a mutual interaction between brain and pelvic organs. Barrington’s nucleus, located in the Pons, plays a key role in this complex mechanism. Barrington’s nucleus neurons project to both pelvic visceral motorneurons (lumbosacral spinal preganglionic neurons giving rise to the parasympathetic innervation of the pelvic viscera) and cerebral norepinephrine neurons (Locus Coeruleus, LC), which modulate behavior. This regulatory circuit coordinates descending limb of the micturition reflex with a central limb that initiates arousal and shifts the focus of attention to facilitate so elimination behavior. It is important to note how elimination disorders can be mutually interconnected with cognitive disorders and behavior. This important association may be due in part to releasing stress-related neuropeptides such as the releasing corticotropin factor (CRF), which, for example, is prominent in the Barrington’s nuclear neurons [7]. CRF is released in PVN and stimulates adrenocorticotropic hormone secretion (ACTH) from the anterior pituitary gland. ACTH stimulates the release of glucocorticoid from the cortex of the adrenal gland [8]. This system is activated under stress conditions, along with another endocrine system, the sympathetic-adrenal-medullar axis, which determines the release of adrenaline and noradrenaline from the adrenal medulla [9,10]. These mechanisms determine metabolic, cardiovascular, immune and behavioral responses [11,12,13]. The gastrointestinal tract from the esophagus to the distal colon is innervated by efferent vagal parasympathetic fibers from the dorsal motor nucleus of the vagus nerve [14]. Studies in animals have shown that vagotomy reduces the motility in the distal colon [14,15]. Therefore, it would be possible to hypothesize that pathways involved in the stress reactions alter the motility of the colon through parasympatic vagal activation, resulting in impairment of evacuation. On the other hand, efferent Barrington’s nucleus fibers send projections to the sacral spinal cord and terminate in the region of the pelvic nerves [16,17]. These pelvic nerves send projections to the colon and rectum [18]. Studies in animals have shown that pelvic nerve activation modulates the motility of the rectum and the distal colon [19,20]. It is of note that 30–50% of all children with fecal incontinence have comorbid emotional or behavioral disorders.

Specifically, data in the literature show that children with fecal incontinence have increased separation anxiety (4.3%), specific phobias (4.3%), general anxiety (3.4%), ADHD (9.2%) and oppositional defiant disorder (11.9%) [21], although relevant stressful events seem not to be involved in the pathogenesis of fecal incontinence [22]. However, data on comorbidities are limited and have mainly focused on ADHD and enuresis [23]. Considering the frequent relationship between enuresis, ADHD and sleep regulation, it is interesting to evaluate possible deficits of sleep organization in FNRFI children [24,25]. Sleep organization disorders have been already associated with the pathogenesis of enuresis. A series of studies, in fact, have shown that the noradrenergic projections from LC are responsible for the arousal, as the LC is activated by stimulation of bladder relaxation during deep sleep [26].

In light of this evidence, one could speculate that the orexinergic system is crucial not only for sleep organization, but also for the sphincteric control. However, data considering sleep organizational disorders in FNRFI children are sparse. The aim of this study was to reduce this gap of knowledge by focusing on specific neurophysiological aspects and Orexin-A plasma levels as measured in FNRFI vs. control children. To the best our knowledge, this is the first polysomnographic/orexin-A association study performed in FNRFI children.

## 2. Materials and Methods

### 2.1. Study Design

The present study and experimental design represented the efforts of a multicenter case-control protocol, involving Sleep Labs at University Child and Adolescent Neuropsychiatry and Paediatrics Clinic in different Regions in Italy (Campania, Umbria, Abruzzo, Calabria, Sicily), and on the Island of Malta.

### 2.2. Population

Forty five (45) children (24 males and 21 females, mean age 9.13 ± 1.31) affected by FNRFI were recruited and compared with 45 typically developing children (TD) (controls) (22 males and 23 females, mean age 9.32 ± 1.5).

Exclusion criteria were: intellectual disability (IQ < 70), autism spectrum disorders (ASD), obesity (z-BMI ≥ 95 percentile) and overweight (z-BMI ≥ 85 percentile), epileptic disorders. Children of both groups were all Caucasians.

The patients were recruited in the same urban area with comparable socio-economic status as computed by Hollingshead Four Factor Index of Social Status [27].

All parents gave their informed consent prior clinical assessment (PSG recordings), plasma orexin-A detection, and for the de-identified dissemination of the results for scientific purposes. This study was carried out in accordance with the principles of the Declaration of Helsinki and was registered in European Clinical Trials Registry (EuDRACT 2015-001163-39).

### 2.3. Polysomnographic Sleep Recordings and Scoring

As described in Roccella et al. [28], sleep macrostructural patterns, nocturnal respiratory events, and periodic limb movements index (PLMI) among FNFRI and TD children were assessed with full polysomnographic recordings (PSG) and visually scored according to the standard criteria of the American Academy of Sleep Medicine (AASM) [29].

An expert scorer (MC) evaluated all the following conventional sleep parameters: Time in bed (TIB), Sleep period time (SPT), Total sleep time (TST), Sleep latency (SOL), REM latency (FRL), Number of stage shifts/hour (SS/h), Number of awakenings/hour (AWN/h), Sleep efficiency (SE%), Percentage of sleep period time spent in sleep stages 1 (N1%) and 2 (N2%), slow-wave sleep (N3%), and REM sleep (REM%), Arousal indexes during the NREM and REM periods.

All variables were analyzed by Hypnolab 1.2 sleep software analysis (SWS Soft, Troina, Italy), according to the international criteria for pediatric age. According to the international criteria, we selected only the last PSG recordings for analysis.

### 2.4. Plasma Orexin-A Detection

Blood samples were drawn from a peripheral vein at 8:00AM after overnight fast into Vacutainer tubes (BD, Franklin Lakes, NJ, USA) containing EDTA and 0.45 TIU/mL of aprotinin. First, each sample was mixed and then immediately a centrifugation at 3000 rpm for 12 min at 4 °C was executed. Plasma was stored at −80 °C until analysis. Enzyme-linked immunoassay kits (ELISA) were used to measure plasma Orexin-A levels. Hypocretin orexin-A1 ELISA kits were purchased from Phoenix Pharmaceuticals Inc. (Burlingame, CA, USA). Before measurements, plasma Orexin-A was extracted using Sep-Pak C18 columns (Waters Corporation, Milford, MA, USA). 10 mL of methanol and 20 mL of H_2_O were used to activate the columns. The following step consisted of adding 1–2 mL of sample to the column and washing with 20 mL of water. Samples were eluted slowly with 80% acetonitrile and resulting volume was reduced to 400 μL under nitrogen flow. An aliquot was led to exsiccation using Speedvac (Savant Instruments, Holbrook, NY, USA). The dry residue was dissolved in water and used for ELISA. There was no cross-reactivity of the antibody for hypocretin-1(16–33), hypocretin-2, agouti-related protein (83–132)-amide. The minimal detectable concentration was 0.37 ng/mL, the intra-assay error <5% and the inter-assay error <14%.

### 2.5. Statistical Analysis

Descriptive statistics were expressed as medians and interquartile ranges (IQR) for continuous variables. Comparisons of categorical data were performed with chi-squared test and Fisher’s exact test, while continuous data were analyzed with nonparametric Mann–Whitney *U* test. A *p* value ≤ 0.05 was considered statistically significant. The software Statistica version 8.1 (StatSoft Inc, Tulsa, OK, USA) was used for all statistical tests.

## 3. Results

No differences between FNRFI and TD children were found for the mean age (t-values = −0.6661; *p* = 0.51) and gender (chi-square = 0.044; *p* = 0.833). Plasma Orexin-A levels in FNFRI children (981.5 ± 252.4 pg/mL; range 729–1223 pg/mL) were significantly higher than TD children (584.3 ± 291.8 pg/mL; range 875–293 pg/mL; t = 6.906, *p* < 0.001) (Table 1).

Table 1 shows differences between FNRFI) vs. TD children groups for the mean age, gender, and plasma Orexin-A levels (pg/mL). Chi-square test and *t*-Students’ analyses were performed where appropriate. *p* ≤ 0.05 values were considered significant.

Table 2 summarizes the PSG parameters between two study groups: FNRFI vs. TD children. Specifically, as for macrostructural sleep stages, FNRFI children showed higher values of sleep onset latency (SOL-min) (*p* < 0.001), stage shifting-h (SS-h) (*p* < 0.001), AWN-h (*p* < 0.001), and lower Time in Bed (TIB-min) (*p* < 0.001), Sleep Partial Time (SPT-min) (*p* < 0.001), Total Sleep Time (TST-min) (*p* < 0.001), Sleep Efficiency percentage (SE%) (*p* < 0.001), stage 2 percentage (N2%) (*p* < 0.001)vs. TD children. Moreover differences were found also for the arousal index in both NREM (*p* < 0.001) and REM (*p* = 0.013) sleep arousal index.

Table 2 shows the differences between FNRFI vs. TD children for the following polysomnographic (PSG) sleep parameters: Time in bed (TIB), Sleep period time (SPT), Total sleep time (TST), Sleep latency (SOL), first REM latency (FRL), stage shifts/hour (SS-h), number of awakenings/hour (AWN/h), Sleep efficiency% (SE%), Percentage of each sleep stage (N1%; N2%, N3%, REM%),NREM and REM arousal index; apnea/hypopnea index (AHI), Oxygen desaturation index (ODI), SpO_2_%, SpO_2_ nadir percentage, SpO_2_ desaturation, Periodic Limb Movement Index (PLMI). t-Student’s analysis was performed. *P* values < 0.05 were considered as significant.

As for the respiratory nocturnal parameters, FNRFI children showed higher apnea/hypopnea/hour (AHI) (*p* < 0.0001), oxygen desaturation index (ODI), lower SpO_2_% (*p* < 0.0001) and nadir SpO_2_% (*p* < 0.0001) and the Periodic Limb Movements Index/hour (PLMI) (*p* < 0.0001) than TD children (Table 2).

## 4. Discussion

Our findings show that FNRFI children have a relevant reduction in sleep stages representation, mainly resumed in the lack of adequate sleep duration and reducing noteworthy sleep efficiency. Particularly, stage 2 and REM sleep% are reduced in FNRI children while between 9 and 16 years stage 2 sleep tend to be more represented, corresponding to the decline in synaptic connectivity among neurons during the transition towards adulthood Evacuation disorders are not just incontinence or bedwetting, but they are indeed a more complex clinical entity that still needs to be fully considered and studied in depth. Evacuation disorders in pediatric age pose a physical, social, and emotional burden for children and families. Indeed, children with FNRFI have been described as having more symptoms of anxiety/depression, family environment with less expressiveness and poor organization, greater attention difficulties, more social problems, more destructive behaviors and school-based school performance [30]. These typical aspects may be explained also by the sleep macrostructural parameters reduction, considering the relevance of adequate sleep efficiency in neurodevelopmental disorders.

In particular, studies have shown that children with primary nocturnal monosymptomatic enuresis (PMNE) and FNRFI present a higher rate of daytime incontinence and micturition problems, thickened bladder walls, pathological electroencephalography, increased frequency of hyperkinetic syndromes, emotional disturbances (anxiety/depression) and behavioral disorders [31]. The higher rate in sleep discontinuity (awakenings/h, stage shifting/h, wake after sleep onset %), the sleep-related breathing disorders and the pathological index for periodic limb movements may sustain the diurnal hyperactivity, behavioral and emotional disorders [32,33,34].

It seems then possible that PMNE and FNRI may be somehow connected to each other starting from their related neuroanatomical circuits and corresponding regulatory mechanisms. For example, the Barrington’s pontine nucleus regulates both micturition and activity of other pelvic organs such as colon and genitals and these type of neurons are activated when bladder pressure increases and receives afferent information from the bladder, spinal cord, and periaqueductal gray region, relevant in modulating stress effects on micturition regulation. The Barrington’s nuclear neurons have also synaptic contacts with the distal colon [35] and this could suggest that these systems may function in a reciprocal way as evidenced by co-activation of bowel and bladder during stressful conditions [36]. In addition, the Barrington’s nuclear neurons projecting to the spinal cord also project to the locus coeruleus (LC) [37,38,39] that projecting to the forebrain, tend to play a key role in activating and maintaining excitement in response to stimuli [38,40]. LC neurons are mainly triggered by visceral stimuli such as bladder and colon distension [41,42,43]. The activation of LC in response to intestinal and pelvic stimuli is correlated with cortical electroencephalographic indices of excitement, such as desynchronization and a shift from high-rise, low-frequency activity at low intensity, high-frequency activity [44,45,46].

In general, alterations of the complex system involving the Barrington’s nucleus may sustain pathophysiological mechanisms underlying elimination disorders, considering the frequent association between fecal incontinence, enuresis and sleep disturbances [47]. On the other hand, LC is associated with the hypocretin/orexin system playing a key role in sleep/wake regulation and the administration of hypocretins directly in LC tend to increase wakefulness and to reduce REM sleep [48]. Moreover, the orexin receptors are expressed in Barrington’s nucleus [49] supporting the role of the orexins system in sleep alterations of FNFRI children as described in our study and reported in other neurodevelopmental disorders [50].

In the medical literature, there are currently very limited data reporting sleep quality in children with fecal incontinence. Considering the social impact of this disorder, a better understanding of the pathophysiological mechanisms underlying this disturb is crucial.

We have taken into account some limitations of the present study such as lack of follow-up evaluation, which could represents a second step of the current study. However, we also would like to emphasize one of the strengths of this study that consisted in the relatively large number of FNFRI children enrolled and that this is one of the first report describing alterations of sleep PSG and Orexin-A abnormalities in these pediatric patients. Also, we propose that the analysis of sleep habits, PSG recordings and Orexin-A plasma levels, may be considered as aclinical tool for the screening and confirmation of FNFRI in children. However, further studies are necessary to establish specificity and sensitivity of the polysomnographic recordings and their orexin-system plasma correlates in the context of a complex disorder of the development, which remains still to be better understood.

## Figures and Tables

**Table 1 brainsci-10-00129-t001:** Characteristics of participants (mean ± SD).

	FNRF IN = 45	TD N = 45	*p*
**Age**	9.13 ± 1.21	9.32 ± 1.5	0.51
**Gender (M/F)**	24/21	22/23	0.833
**Plasma Orexin-A levels (pg/mL)**	981.5 ± 252.4	584.3 ± 291.8	<0.001

**Table 2 brainsci-10-00129-t002:** Polysomnographic macrostructural parameters comparison between the groups (median, inter quartriles analysis; Mann-Whitney U test).

	FNRFI Children N = 45		TD Children N = 45			
	Median	IQR	Median	IQR	z-Value	*p*
**Time in Bed (TIB)-min**	474.35	434.617–513.45	573	506–609	−6.363	<0.0001
**Sleep Partial Time (SPT)-min**	409	382–474.3	544.5	485–596	−6.532	<0.0001
**Total Sleep Time (TST)-min**	352.7	325.5–411.1	518	471–552	−7.719	<0.0001
**Sleep Onset Latency (SOL)-min**	39.183	18.8–76.3	19.5	9.5–29	3.611	<0.0001
**First REM Latency (FRL)-min**	100.5	98.5–125.5	129.5	92.5–152	−1.521	NS
**Stage Shifting (SS)-h**	10.2	9.2–10.7	5.8	4.8–7.8	5.415	<0.0001
**Awakenings (AWN)-h**	8.4	4.3–10.4	1.2	0.4–2.2	6.766	<0.0001
**Sleep Efficiency (SE)%**	76.538	69.593–81.382	90.6	87.2–94.5	−6.452	<0.0001
**Wake After Sleep Onset (WASO)%**	14.903	12.338–18.454	3.7	0.6–7.4	6.403	<0.0001
**Stage 1 (N1)%**	10.436	5.116–15.564	1.6	1–4	5.701	<0.0001
**Stage 2 (N2)%**	28.629	22.648–32.689	43.4	35.2–47.7	−5.645	<0.0001
**Slow waves sleep (N3)%**	27	16.933–29.609	27.6	23.8–34.8	−1.650	NS
**Rapid eyes movement (REM)%**	19.969	17.146–23.235	21.5	16.2–24.1	−0.238	NS
**NREM Arousal Index**	9.7	9.2–10.2	8.0	6.6–8.6	6.540	<0.0001
**REM Arousal Index**	8.8	8.3–9.4	9.4	8.7–9.8	−2.494	0.013
**Apnea/Hypopnea (AHI)/h**	6.4	5.2–8.1	0.3	0.2–06	8.171	<0.0001
**Oxygen desaturation index (ODI)**	3.9	2.6–4.8	0.0	0.0–0.3	8.171	<0.0001
**Oxygen Saturation (SpO_2_)%**	97.6	96.8–98.1	98.7	98.4–98.8	−6.064	<0.0001
**Oxygen saturation (SpO_2_) nadir%**	92	89.4–94.4	98	97.5–98	−7.767	<0.0001
**Oxygen desaturation (SpO_2_) %**	3.9	3.2–5.3	0.0	0.0–1.1	7.856	<0.0001
**Periodic Limb Movements Index (PLMI)**	5.12	3.7–6.42	3.1	2.1–3.4	5.838	<0.0001

NS: Not significant.

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
