# Peer review of "Neuropeptides’ Hypothalamic Regulation of Sleep Control in Children Affected by Functional Non-Retentive Fecal Incontinence"

_brainsci, 2020, doi:10.3390/brainsci10030129_

Round 1

Reviewer 1 Report

The authors present an interesting study in a field where not much research has been performed.

Extensive editing of English language and style is required before this paper can be considered for publication. I have not taken the time to point out all grammatical errors, only the wrong use of some specific words/terms since these should not be overlooked when revising this paper.

Terminology: Please refrain from using the word "encopresis", this term has been abandoned a long time ago and may cause confusion since some may think of it as a term for constipation-associated feral incontinence. The authors have used functional non-retentive fecal soiling, however this is also not the terminology used in the Rome publications or ESPGHAN/NASPGHAN papers. Instead, please only use the term "fecal incontinence" throughout the entire manuscript.

Introduction:

The authors refer to the definition by the DSM but for epidemiological and other data to Rome publications. This is very confusing since psychiatrists and pediatric gastroenterologists approach this problem from a different angle and do not always use the same definitions. I strongly suggest to adhere to the Rome IV definitions, these are the most recent and the most well-accepted. Please refer to the Rome criteria and mention them.

A few important spelling errors and wrong usage of words are listed here. The authors use "bark of the adrenal gland", this should be "cortex", "from the midrange of the adrenaline" should be "adrenal medulla", "animal shave" should be "animals have", "is scares" should be "is scarce".

Methods:

"FNFRS" (L139) should be "FNRFS" or in my opinion "FNRFI" because soiling should be replaced by incontinence.

2.3: Please do not write out the complete references in the text, but use them as you did before, as references. 

2.4: I have never encountered the term "forearm vein", please rephrase.

Some elements of the methods could be published as an appendix, since it is not very relevant to the general reader. The majority of section 2.4 is an example.

2.5: The authors have chosen to use a statistical analysis for normally distributed data. I strongly doubt the data were indeed normally distributed with such small sample sizes, if the data are not normally distributed a different test should have been used (Mann Whitney U). This is very important for the entire paper. Moreover, if data are not normally distributed, I suggest reporting the median, IQR and range rather than the mean and SD.

Results

Taking into account the above comment on the statistics, it is difficult to interpret the results of the student T tests. 

The tables and text are full of abbreviations, making it impossible to easily read through and comprehend the results without constantly looking at the meaning of the abbreviations. A further explanation of the results is required here.

The authors state to have performed Pearson correlation tests, but I do not see these reported?

Discussion

The discussion contains quite a lot of repetition. On the other hand, the authors hardly discuss the actual main findings of the study: the association between abnormal sleep and FNRFI. This deserves more attention.

Author Response

Reviewer 1

Comments and Suggestions for Authors

The authors present an interesting study in a field where not much research has been performed.

Extensive editing of English language and style is required before this paper can be considered for publication. I have not taken the time to point out all grammatical errors, only the wrong use of some specific words/terms since these should not be overlooked when revising this paper.

Replay

Thank you for your comment. I improved the English language as suggested

Reviewer 1

Terminology: Please refrain from using the word "encopresis", this term has been abandoned a long time ago and may cause confusion since some may think of it as a term for constipation-associated feral incontinence. The authors have used functional non-retentive fecal soiling, however this is also not the terminology used in the Rome publications or ESPGHAN/NASPGHAN papers. Instead, please only use the term "fecal incontinence" throughout the entire manuscript.

Replay

Thank you for your comment. I modified in all the text the term  as suggested.

Reviewer 1

Introduction:

The authors refer to the definition by the DSM but for epidemiological and other data to Rome publications. This is very confusing since psychiatrists and pediatric gastroenterologists approach this problem from a different angle and do not always use the same definitions. I strongly suggest to adhere to the Rome IV definitions, these are the most recent and the most well-accepted. Please refer to the Rome criteria and mention them.

Replay

Thank you for your comment. I have modified the text as follow:

Functional non-retentive fecal soiling (FNRFI) is a relatively common problem in pediatric age [1 Philichi, 2008. FNRFI is associated with high levels of distress for both children and parents and with emotional disorders in about 30%-50% of affected children [2 Von Gontard et al., 2011]. Two major forms of fecal incontinence can be differentiated: fecal incontinence with and without constipation. According to the Rome IV criteria [3 Drossman L] fecal incontinence is defined as unintended loss of solid or liquid stool occurring in inappropriate places in a 4-year-old or older child after organic causes have been ruled out.

Although epidemiological data are still contrasting and the estimated prevalence depending on the definition used, FNRFI tends to affects up to 3% of children older than 4 years [4 Rome Foundation Guidelines, 2006; 5 Von Gontard, 2012]. Moreover, males seem to be affected more frequently than females and daytime FNRFI is more frequent than nocturnal FNRFI, which is most often due to organic causes [6 Joinson, 2006].

Clinically, children affected by FNRFI have normal intestinal movements and stool consistency, infrequent abdominal pain, normal colon transit and no stool mass. Enuresis and urinary incontinence are less common as comorbidity. In general, coordination of pelvic visceral activities allowing appropriate elimination behaviors requires a mutual interaction between brain and pelvic organs. Barrington's nucleus, located in the Pons, plays a key role in this complex mechanism. Barrington's nucleus neurons project to both pelvic visceral motorneurons (lumbosacral spinal preganglionic neurons giving rise to the parasympathetic innervation of the pelvic viscera) and cerebral norepinephrine neurons (Locus Coeruleus, LC), which modulate behavior. This regulatory circuit coordinates descending limb of the micturition reflex with a central limb that initiates arousal and shifts the focus of attention to facilitate so elimination behavior. It is important to note how elimination disorders can be mutually interconnected with cognitive disorders and behavior. This important association may be due in part to releasing stress-related neuropeptides such as the releasing corticotropin factor (CRF), which, for example, is prominent in the Barrington's nuclear neurons [7 Bellman, 1966]. CRF is released in PVN and stimulates adrenocorticotropic hormone secretion (ACTH) from the anterior pituitary gland. ACTH stimulates the release of glucocorticoid from the cortex of the adrenal gland [8 Valentino et al., 2011]. This system is activated under stress conditions, along with another endocrine system, the sympathetic-adrenal-medullar axis, which determines the release of adrenaline and noradrenaline from the adrenal medulla [9 Lightman, 2008; Petito et al., 2016; 10 Bertozzi et al., 2017]. These mechanisms determine metabolic, cardiovascular, immune and behavioral responses [11 Buske-Kirschbaum et al., 2002; 12 Charmandari et al., 2005; 13 Yang and Glaser, 2000]. The gastrointestinal tract from the esophagus to the distal colon is innervated by efferent vagal parasympathetic fibers from the dorsal motor nucleus of the vagus nerve [14 De Vente et al., 2003]. Studies in animals have shown that vagotomy reduces the motility in the distal colon [14 Berthoud et al., 1991; 15 Lenz, 1989]. Therefore, it would be possible to hypothesize that pathways involved in the stress reactions alter the motility of the colon through parasympatic vagal activation, resulting in impairment of evacuation. On the other hand, efferent Barrington’s nucleus fibers send projections to the sacral spinal cord and terminate in the region of the pelvic nerves [16 Nakade et al., 2007; 17 Loewy, 1979]. These pelvic efferents send colon and rectum projections [18 Blok and Holstege, 1997]. Studies in animals have shown that pelvic nerve activation modulates the motility of the rectum and the distal colon [19 Luckensmeyer and Keast, 1998; 20 Ridolfi et al., 2009]. In this context, it is noteworthy that 30-50% of all children with fecal incontinence have comorbid emotional or behavioral disorders.

Specifically, data in the literature show that children with fecal incontinence have an increased levels of separation anxiety (4.3%), specific phobias (4.3%), general anxiety (3.4%), ADHD (9.2%) and oppositional defiant disorder (11.9%) [21 Suda et al., 2013], although no relevant stressful events seem to be involved in fecal incontinence pathogenesis [22 Ruberto et al., 2019]. However, data on comorbidities are limited and have mainly focused on ADHD and enuresis [23 Joinson et al., 2006]. Considering the frequent relationship between enuresis, ADHD and sleep regulation reported by clinical literature, it appears suggestive and intriguing to evaluate possible deficits of sleep organization in FNRFI children [24 Mellon et al., 2013; 25 Spruyt and Gozal, 2011]. Sleep organization disorders have been already observed in the pathogenesis of enuresis. A series of studies, in fact, have shown that the noradrenergic projections from LC are responsible for the arousal, and the LC is indeed activated by stimulation of bladder relaxation during deep sleep [26 Kanata et al., 2016].

At the light of these studies, we could also speculate that the orexinergic system is crucial not only for sleep organization but also for the sphincteric control. However, the medical literature considering sleep organizational disorders in FNRFI children is scarce. The aim of this study was to reduce this gap of knowledge by focusing on specific neurophysiological aspects and Orexin-A plasma levels as measured in FNRFI vs. control children. At the best our knowledge, this is the first polysomnographic and Orexin-A correlation study performed in FNRFI children.

Reviewer 1

A few important spelling errors and wrong usage of words are listed here. The authors use "bark of the adrenal gland", this should be "cortex", "from the midrange of the adrenaline" should be "adrenal medulla", "animal shave" should be "animals have", "is scares" should be "is scarce".

Replay

Thank you for your comment. I have modified the text as suggested.

Reviewer 1

Methods:

"FNFRS" (L139) should be "FNRFS" or in my opinion "FNRFI" because soiling should be replaced by incontinence.

2.3: Please do not write out the complete references in the text, but use them as you did before, as references. 

2.4: I have never encountered the term "forearm vein", please rephrase.

Some elements of the methods could be published as an appendix, since it is not very relevant to the general reader. The majority of section 2.4 is an example.

Replay

Thank you for your comment. I have modified the text as suggested.

Materials and Methods

2.1. Study design

The present study and experimental design represented the efforts of a multicenter case-control protocol, involving Sleep Labs at University Child and Adolescent Neuropsychiatry and Paediatrics Clinic in different Regions in Italy (Campania, Umbria, Abruzzo, Calabria, Sicily), and in Malta Isle.

2.2. Population

45 children (24 males and 21 females, mean age 9.13±1.31) affected by FNRFI were recruited and compared with 45 typically developing children (TD) (controls) (22 males and 23 females, mean age 9.32 ±1.5).

Exclusion criteria were: intellectual disability (IQ<70), autism spectrum disorders (ASD), obesity (z-BMI ≥95 percentile) and overweight (z-BMI ≥85 percentile), epileptic disorders. Children of both groups were all Caucasians.

They were recruited in the same urban area with comparable socio-economic status as computed by Hollingshead Four Factor Index of Social Status [27 Hollingshead, 1975].

All parents gave their informed consent prior clinical assessment (PSG recordings), plasma orexin-A detection, and for the deidentified dissemination of the results for scientific purposes. This study was carried out in accordance with the principles of the Declaration of Helsinki and was registered in European Clinical Trials Registry (EuDRACT 2015-001163-39).

2.3. Polysomnographic sleep recordings and scoring

As described in Roccella et al. [28 Roccella M, et al. 2019], sleep macrostructural patterns, nocturnal respiratory events, and periodic limb movements index (PLMI) among FNFRI and TD children were assessed with full polysomnographic recordings (PSG) and visually scored according to the standard criteria of the American Academy of Sleep Medicine (AASM) [29 Iber et al., 2007].

An expert scorer (MC) evaluated all the following conventional sleep parameters: Time in bed (TIB), Sleep period time (SPT), Total sleep time (TST), Sleep latency (SOL), REM latency (FRL), Number of stage shifts/hour (SS/h), Number of awakenings/hour (AWN/h), Sleep efficiency (SE%), Percentage of sleep period time spent in sleep stages 1 (N1%) and 2 (N2%), slow-wave sleep (N3%), and REM sleep (REM%), Arousal indexes during the NREM and REM periods.

All variables were analyzed by Hypnolab 1.2 sleep software analysis (SWS Soft, Troina, Italy), according to the international criteria for pediatric age. According to the international criteria, we selected only the last PSG recordings for analysis.

2.4. Plasma Orexin-A detection

Blood samples were drawn from a peripheral vein at 8:00AM after overnight fast into Vacutainer tubes (BD, Franklin Lakes, NJ, USA) containing EDTA and 0.45 TIU/ml of aprotinin. First, each sample was mixed and then immediately a centrifugation at 3000 rpm for 12 min at 4°C was executed. Plasma was stored at −80°C until analysis. Enzyme-linked immunoassay kits (ELISA) were used to measure plasma Orexin-A levels. Hypocretin Orexin-A1 ELISA kits were purchased from Phoenix Pharmaceuticals (Phoenix Pharmaceuticals Inc., CA, USA). Before measurements, plasma Orexin-A was extracted using Sep-Pak C18 columns (Waters Corporation, Milford, MA, USA). 10 ml of methanol and 20 ml of H2O were used to activate the columns. The following step consisted of adding 1-2 ml of sample to the column and washing with 20 ml of water. Samples were eluted slowly with 80% acetonitrile and resulting volume was reduced to 400μl under nitrogen flow. An aliquot was led to exsiccation using Speedvac (Savant Instruments, Holbrook, NY, USA). The dry residue was dissolved in water and used for ELISA. There was no cross-reactivity of the antibody for hypocretin-1(16–33), hypocretin-2, agouti-related protein (83–132)-amide. The minimal detectable concentration was 0.37 ng/ml, the intra-assay error <5% and the inter-assay error <14%.

Reviewer 1

2.5: The authors have chosen to use a statistical analysis for normally distributed data. I strongly doubt the data were indeed normally distributed with such small sample sizes, if the data are not normally distributed a different test should have been used (Mann Whitney U). This is very important for the entire paper. Moreover, if data are not normally distributed, I suggest reporting the median, IQR and range rather than the mean and SD.

Replay

I heve modified the text as follow:

2.5. Statistical analysis

Chi-square and t-Student’s analyses were performed when appropriated to compare the two populations (FNRFI and TD children) for age, gender, full polysomnographic parameters, and Orexin-A plasma levels. Pearson’s analysis was performed to verify the correlation between sleep parameters and Orexin-A plasma levels in both groups.

Reviewer 1

Results

Taking into account the above comment on the statistics, it is difficult to interpret the results of the student T tests. 

The tables and text are full of abbreviations, making it impossible to easily read through and comprehend the results without constantly looking at the meaning of the abbreviations. A further explanation of the results is required here.

Replay

Thank you for your comment. I have modified the manuscript as suggested.

. Results

No differences between FNRFI and TD children were found for the mean age (t-values= -0.6661; p=0.51) and gender (chi-square= 0.044; p=0.833). Plasma Orexin-A levels in FNFRI children (981.5± 252.4 pg/ml; range 729–1223 pg/ml) were significantly higher than TD children (584.3±291.8 pg/ml; range 875–293 pg/ml; t= 6.906, p<0.001) (Table 1).

Table 1.

FNRFI

N=45

TD

N=45

p

Age

9.13±1.21

9.32 ±1.5

0.51

Gender (M/F)

24/21

22/23

0.833

Plasma Orexin-A levels (pg/ml)

981.5 ± 252.4

584.3±291.8

<0.001

Table 1 shows differences between FNRFI) vs. TD children groups for the mean age, gender, and plasma Orexin-A levels (pg/ml). Chi-square test and t-Students’ analyses were performed where appropriate. p≤0.05 values were considered significant.

Table 2 summarizes the PSG parameters between two study groups: FNRFI vs. TD children. Specifically, as for macrostructural sleep stages, FNRFI children showed higher values of SOL-min (p<0.001), SS-h (p<0.001), AWN-h (p<0.001), and lower TIB-min (p<0.001), SPT-min (p<0.001), TST-min (p<0.001), SE% (p<0.001), N2-spt (p<0.001), N3-spt (p<0.001) vs. TD children. Moreover differences were found also for the arousal index in both NREM (p<0.001) and REM (p=0.028) sleep arousal index.

Table 2.

FNRFI Children

N=45

TD Children

N=45

Mean

SD

Mean

SD

t-value (df=88)

p

TIB-min

469.55

51.57

583.36

88.38

-7.46

<0.001

SPT-min

417.20

70.97

551.19

79.03

-8.46

<0.001

TST-min

356.24

71.05

525.81

71.43

-11.29

<0.001

SOL-min

52.25

42.79

22.67

18.05

4.27

<0.001

FRL-min

116.92

38.67

129.83

50.35

-1.36

NS

SS-h

9.92

2.99

6.44

2.15

6.35

<0.001

AWN-h

7.69

3.80

1.72

1.77

9.54

<0.001

SE%

75.60

11.08

90.48

5.43

-8.09

<0.001

WASO-spt

10.61

6.38

3.37

4.28

6.32

0.000

N1%

28.92

9.85

41.92

8.12

-6.83

0.000

N2%

24.54

8.57

29.06

10.32

-2.26

0.026

N3%

10.61

6.38

3.37

4.28

6.32

0.000

REM%

21.07

5.29

21.21

5.56

-0.12

NS

NREM Arousal Index

9.73

0.94

7.60

1.41

8.45

0.000

REM Arousal Index

8.86

0.82

9.29

1.02

-2.23

0.028

AHI

7.12

2.49

0.42

0.26

17.91

<0.001

ODI

3.79

1.79

0.13

0.22

13.61

<0.001

SpO2%

97.14

1.43

98.55

0.36

-6.40

<0.001

SpO2 nadir%

91.61

4.46

97.79

0.55

-9.21

<0.001

SpO2 desat%

4.13

1.96

0.49

0.57

11.95

<0.001

PLMI

5.16

1.70

2.90

0.97

7.73

<0.001

Table 2 shows the differences between FNRFI vs. TD children for the following polysomnographic (PSG) sleep parameters: Time in bed (TIB), Sleep period time (SPT), Total sleep time (TST), Sleep latency (SOL), first REM latency (FRL), stage shifts/hour (SS-h), number of awakenings/hour (AWN/h), Sleep efficiency% (SE%), Percentage of each sleep stage (N1%; N2%, N3%, REM%),NREM and REM arousal index;  apnea/hypopnea index (AHI), Oxygen desaturation index (ODI), SpO2%, SpO2 nadir percentage, SpO2 desaturation, Periodic Limb Movement Index (PLMI). t-Student’s analysis was performed. P values <0.05 were considered as significant.

As for the respiratory nocturnal parameters, FNRFI children showed higher apnea/hypopnea/hour (AHI) (p<0.001), oxygen desaturation index (ODI), lower SpO2% (p<0.001) and SpO2 nadir% (p<0.001) and the Periodic Limb Movements Index/hour (PLMI) (p<0.001) than TD children (Table 2).

Reviewer 1

Discussion

The discussion contains quite a lot of repetition. On the other hand, the authors hardly discuss the actual main findings of the study: the association between abnormal sleep and FNRFI. This deserves more attention.

Replay

Thak you for your comment. I have modified the text as follow:

Discussion

Our findings show that FNRFI children have a relevant reduction in sleep stages representation, mainly resumed in the lack of adequate sleep duration and reducing noteworthy sleep efficiency. Evacuation disorders are not just incontinence or bedwetting, but they are indeed a more complex clinical entity that still needs to be fully considered and studied in depth. Evacuation disorders in pediatric age pose a physical, social, and emotional burden for children and families. Indeed, children with fecal incontinence have been described as having more symptoms of anxiety/depression, family environment with less expressiveness and poor organization, greater attention difficulties, more social problems, more destructive behaviors and school-based school performance [30 Watanabe and Kawauchi, 1999].

In particular, studies have shown that children with enuresis and fecal incontinence present a higher rate of daytime incontinence and micturition problems, thickened bladder walls, pathological electroencephalography, increased frequency of hyperkinetic syndromes, emotional disturbances (anxiety/depression) and behavioral disorders [31 Cox et al., 2002]. It seems then possible that these two disabilities may be somehow connected to each other starting from their related neuroanatomical circuits and corresponding regulatory mechanisms. For example, the Barrington's pontine nucleus regulates both micturition and activity of other pelvic organs such as colon and genitals. Moreover, Barrington's nuclear neurons are activated when bladder pressure increases and receives afferent information from the bladder, spinal cord, and periaqueductal gray region, which has an important role in modulating stress effects on micturition regulation. The Barrington's nuclear neurons have also synaptic contacts with the distal colon [32 Von Gontard and Hollmann, 2004] and this could suggest that these systems may function in a reciprocal way. For example, in conditions like stress, there is co-activation of bowel and bladder and information coming from both organs converge on Barrington's nuclear neurons [33 Rouzade-Dominguez et al., 2003a]. In addition, the Barrington's nuclear neurons projecting to the spinal cord also project to LC, the main source of norepinephrine in the brain [34 Rouzade-Dominguez et al., 2003b; 35 Aston-Jones et al., 1995; 36 Swanson and Hartman, 1975]. The LC contains neurons that project to the whole neuraxis (especially to the forebrain) and play a key role in activating and maintaining excitement in response to stimuli [37 Aston-Jones and Cohen, 2005; 38 Berridge and Waterhouse, 2003]. LC neurons are mainly triggered by visceral stimuli such as bladder and colon distension [39 Elam et al., 1986; 40 Foote et al., 1980; 41 Lechner et al., 1997]. The activation of LC in response to intestinal and pelvic stimuli is correlated with cortical electroencephalographic indices of excitement, such as desynchronization and a shift from high-rise, low-frequency activity at low intensity, high-frequency activity [42 Page et al., 1992]. The Barrington's nucleus neurons with their projections to the spinal cord and LC integrate then signals from multiple pelvic organs and coordinates their activity with the activity in prosencephalous regions underlying behavior. CRF is indeed expressed in Barrington's nucleus neurons. Specifically, CRF axons from Barrington's nucleus end both to LC and parasympathetic preganglionic nucleus, which innervates the pelvic region. Physiologically, Barrington’s nucleus neurons expressing CFR respond to both bladder and colon distension. Through the projections of the Barrington’s nucleus to the LC, CRF has mainly excitatory effects and modulates neuronal activation in LC in response to pelvic visceral stimuli which, in turn, arouse cortical excitement [43 Lechner et al., 1997]. On the other hand, through the spinal projections from the Barrington’s nucleus, CRF have excitatory effects on colon motoneurons, helping to increase the motility of colon [44 Bellman, 1966].

In general then, it is conceivable that alterations of the complex system involving the Barrington’s nucleus might be involved in specific pathophysiological mechanisms underlying elimination disorders such as enuresis and fecal incontinence. Considering the frequent association between fecal incontinence, enuresis and sleep disturbances, it is interesting to note that LC is associated with the hypocretin/orexin system, which plays indeed a key role in sleep/wake regulation. In support of the anatomo-functional connection between LC and hypocretin/orexin system it has been shown that administration of hypocretins directly in LC increases wakefulness and reduces REM sleep [45 Bourgin et al., 2000].

Conversely, in Barrington’s nucleus, Orexin receptors are expressed as described by immunohistochemistry analysis [46 Greco and Shiromani, 2001] supporting a possible relevant role of the orexins system in sleep alterations of FNFRI children as described in our study.

In the medical literature, there are currently very limited amounts of data reporting sleep quality in children with fecal incontinence. Considering the social impact of this disorder, a better understanding of the pathophysiological mechanisms underlying this disturb is crucial. In this regard, further investigations are needed to improve possible preventive tools and expand therapeutic strategies.

We have taken into account some limitations of the present study such as lack of follow-up evaluation, which could represents indeed a second step of the current study. However, we also would like to emphasize one of the points of strength of this report that consisted in the relatively large number of FNFRI children enrolled and that this is one of the first report describing alterations of sleep PSG and Orexin-A abnormalities in this type of pediatric patients. Also, we propose that the analysis of sleep habits, PSG recordings and Orexin-A plasma levels, may be considered as an affordable clinical tool for the screening and confirmation of FNFRI in children. However, further studies are necessary to establish specificity and sensitivity of the PSG recordings and their orexin-system plasma correlates in the context of a complex disorder of the development, which remains still to be better understood.

Reviewer 2 Report

Reviewer’s comments and suggestions for Authors

This study, focusing on neurophysiological aspects, tries to clarify the relationship between sleep organizational disorders and FNRFS. Furthermore, the main aim was to evaluate the orexin serum levels in children with Functional non-retentive fecal soiling (FNRFS) and secondarily assessing the polysomnographic sleep macrostructure. The authors took two study groups (FNFRS group vs control group), the sleep patterns and respiratory events were measured with polysomnographic recordings (PSG); moreover, they also examined the level of Orexin-A plasma level. Their finding suggests that sleep habits evaluation could be considered as screening and complementary tools for the diagnosis of encopresis in children.

The manuscript is well written in terms of the English language. However, there was some inconsistency in the introduction and discussion part which needed to clear out in the revised version. The topic is of great interest to clinicians working in the area of sleep biology and neurology.

Nonetheless, the manuscript required to improve by reviewing a few minor concerns.

(1) Line 41-42 (in abstract) needed to rewrite or delete because of the inconsistency in the sentences.

(2) Line number 46 (abstract) the author can mention the age group and number of participants

(3) No need to add the line at the ends of abstract “To establish specificity and sensitivity capacity of the polysomnographic approach in the context of a probably much more complex disorder of the development, further studies are necessary” delete it

(4) Line 88-89, the authors need to write which mechanism they were discussing so that the common reader may understand the storyline.

(5) Line 97, have was written as “shave”

(6) Line 145, what was sleep 328 ?

(7) Line 147-148 please check the reference guideline of the journal. Why the authors wrote the reference here.

(8) In statistical analysis, differentiate where the author used chi and t student analysis

(9) In table 1 it would be better to write the title not using N

(10) Line 187, the Control should be control

(11) Line 189, for representing the figures the author needs to write the sentence in the present tense.

(12) Line 211-217, the first paragraph needs to discuss based on the current study result.

(13) Line 223 to 226, Need a reference for the sentences

(14) Line 263-264 “Some limitations discussed here but I could not find whole details of that. The need to be specific about the information that they discussed, they have to consistent about the information presenting in the sentences”.

(15) Line 269-270, the same line written in the abstract, no need to add this information

(16) Please check the references guideline to denote the references. I found mistakes in several references like 12,18,19,20,26,28,29,32,35,40,42 and 43

Author Response

Reviewer 2

Comments and Suggestions for Authors

Reviewer’s comments and suggestions for Authors

 This study, focusing on neurophysiological aspects, tries to clarify the relationship between sleep organizational disorders and FNRFS. Furthermore, the main aim was to evaluate the orexin serum levels in children with Functional non-retentive fecal soiling (FNRFS) and secondarily assessing the polysomnographic sleep macrostructure. The authors took two study groups (FNFRS group vs control group), the sleep patterns and respiratory events were measured with polysomnographic recordings (PSG); moreover, they also examined the level of Orexin-A plasma level. Their finding suggests that sleep habits evaluation could be considered as screening and complementary tools for the diagnosis of encopresis in children.

 The manuscript is well written in terms of the English language. However, there was some inconsistency in the introduction and discussion part which needed to clear out in the revised version. The topic is of great interest to clinicians working in the area of sleep biology and neurology.

Replay

Thak you for your comment. I have modified the text as follow:

Introduction

Functional non-retentive fecal soiling (FNRFI) is a relatively common problem in pediatric age [1 Philichi, 2008. FNRFI is associated with high levels of distress for both children and parents and with emotional disorders in about 30%-50% of affected children [2 Von Gontard et al., 2011]. Two major forms of fecal incontinence can be differentiated: fecal incontinence with and without constipation. According to the Rome IV criteria [3 Drossman L] fecal incontinence is defined as unintended loss of solid or liquid stool occurring in inappropriate places in a 4-year-old or older child after organic causes have been ruled out.

Although epidemiological data are still contrasting and the estimated prevalence depending on the definition used, FNRFI tends to affects up to 3% of children older than 4 years [4 Rome Foundation Guidelines, 2006; 5 Von Gontard, 2012]. Moreover, males seem to be affected more frequently than females and daytime FNRFI is more frequent than nocturnal FNRFI, which is most often due to organic causes [6 Joinson, 2006].

Clinically, children affected by FNRFI have normal intestinal movements and stool consistency, infrequent abdominal pain, normal colon transit and no stool mass. Enuresis and urinary incontinence are less common as comorbidity. In general, coordination of pelvic visceral activities allowing appropriate elimination behaviors requires a mutual interaction between brain and pelvic organs. Barrington's nucleus, located in the Pons, plays a key role in this complex mechanism. Barrington's nucleus neurons project to both pelvic visceral motorneurons (lumbosacral spinal preganglionic neurons giving rise to the parasympathetic innervation of the pelvic viscera) and cerebral norepinephrine neurons (Locus Coeruleus, LC), which modulate behavior. This regulatory circuit coordinates descending limb of the micturition reflex with a central limb that initiates arousal and shifts the focus of attention to facilitate so elimination behavior. It is important to note how elimination disorders can be mutually interconnected with cognitive disorders and behavior. This important association may be due in part to releasing stress-related neuropeptides such as the releasing corticotropin factor (CRF), which, for example, is prominent in the Barrington's nuclear neurons [7 Bellman, 1966]. CRF is released in PVN and stimulates adrenocorticotropic hormone secretion (ACTH) from the anterior pituitary gland. ACTH stimulates the release of glucocorticoid from the cortex of the adrenal gland [8 Valentino et al., 2011]. This system is activated under stress conditions, along with another endocrine system, the sympathetic-adrenal-medullar axis, which determines the release of adrenaline and noradrenaline from the adrenal medulla [9 Lightman, 2008; Petito et al., 2016; 10 Bertozzi et al., 2017]. These mechanisms determine metabolic, cardiovascular, immune and behavioral responses [11 Buske-Kirschbaum et al., 2002; 12 Charmandari et al., 2005; 13 Yang and Glaser, 2000]. The gastrointestinal tract from the esophagus to the distal colon is innervated by efferent vagal parasympathetic fibers from the dorsal motor nucleus of the vagus nerve [14 De Vente et al., 2003]. Studies in animals have shown that vagotomy reduces the motility in the distal colon [14 Berthoud et al., 1991; 15 Lenz, 1989]. Therefore, it would be possible to hypothesize that pathways involved in the stress reactions alter the motility of the colon through parasympatic vagal activation, resulting in impairment of evacuation. On the other hand, efferent Barrington’s nucleus fibers send projections to the sacral spinal cord and terminate in the region of the pelvic nerves [16 Nakade et al., 2007; 17 Loewy, 1979]. These pelvic efferents send colon and rectum projections [18 Blok and Holstege, 1997]. Studies in animals have shown that pelvic nerve activation modulates the motility of the rectum and the distal colon [19 Luckensmeyer and Keast, 1998; 20 Ridolfi et al., 2009]. In this context, it is noteworthy that 30-50% of all children with fecal incontinence have comorbid emotional or behavioral disorders.

Specifically, data in the literature show that children with fecal incontinence have an increased levels of separation anxiety (4.3%), specific phobias (4.3%), general anxiety (3.4%), ADHD (9.2%) and oppositional defiant disorder (11.9%) [21 Suda et al., 2013], although no relevant stressful events seem to be involved in fecal incontinence pathogenesis [22 Ruberto et al., 2019]. However, data on comorbidities are limited and have mainly focused on ADHD and enuresis [23 Joinson et al., 2006]. Considering the frequent relationship between enuresis, ADHD and sleep regulation reported by clinical literature, it appears suggestive and intriguing to evaluate possible deficits of sleep organization in FNRFI children [24 Mellon et al., 2013; 25 Spruyt and Gozal, 2011]. Sleep organization disorders have been already observed in the pathogenesis of enuresis. A series of studies, in fact, have shown that the noradrenergic projections from LC are responsible for the arousal, and the LC is indeed activated by stimulation of bladder relaxation during deep sleep [26 Kanata et al., 2016].

At the light of these studies, we could also speculate that the orexinergic system is crucial not only for sleep organization but also for the sphincteric control. However, the medical literature considering sleep organizational disorders in FNRFI children is scarce. The aim of this study was to reduce this gap of knowledge by focusing on specific neurophysiological aspects and Orexin-A plasma levels as measured in FNRFI vs. control children. At the best our knowledge, this is the first polysomnographic and Orexin-A correlation study performed in FNRFI children.

Discussion

Our findings show that FNRFI children have a relevant reduction in sleep stages representation, mainly resumed in the lack of adequate sleep duration and reducing noteworthy sleep efficiency. Evacuation disorders are not just incontinence or bedwetting, but they are indeed a more complex clinical entity that still needs to be fully considered and studied in depth. Evacuation disorders in pediatric age pose a physical, social, and emotional burden for children and families. Indeed, children with fecal incontinence have been described as having more symptoms of anxiety/depression, family environment with less expressiveness and poor organization, greater attention difficulties, more social problems, more destructive behaviors and school-based school performance [30 Watanabe and Kawauchi, 1999].

In particular, studies have shown that children with enuresis and fecal incontinence present a higher rate of daytime incontinence and micturition problems, thickened bladder walls, pathological electroencephalography, increased frequency of hyperkinetic syndromes, emotional disturbances (anxiety/depression) and behavioral disorders [31 Cox et al., 2002]. It seems then possible that these two disabilities may be somehow connected to each other starting from their related neuroanatomical circuits and corresponding regulatory mechanisms. For example, the Barrington's pontine nucleus regulates both micturition and activity of other pelvic organs such as colon and genitals. Moreover, Barrington's nuclear neurons are activated when bladder pressure increases and receives afferent information from the bladder, spinal cord, and periaqueductal gray region, which has an important role in modulating stress effects on micturition regulation. The Barrington's nuclear neurons have also synaptic contacts with the distal colon [32 Von Gontard and Hollmann, 2004] and this could suggest that these systems may function in a reciprocal way. For example, in conditions like stress, there is co-activation of bowel and bladder and information coming from both organs converge on Barrington's nuclear neurons [33 Rouzade-Dominguez et al., 2003a]. In addition, the Barrington's nuclear neurons projecting to the spinal cord also project to LC, the main source of norepinephrine in the brain [34 Rouzade-Dominguez et al., 2003b; 35 Aston-Jones et al., 1995; 36 Swanson and Hartman, 1975]. The LC contains neurons that project to the whole neuraxis (especially to the forebrain) and play a key role in activating and maintaining excitement in response to stimuli [37 Aston-Jones and Cohen, 2005; 38 Berridge and Waterhouse, 2003]. LC neurons are mainly triggered by visceral stimuli such as bladder and colon distension [39 Elam et al., 1986; 40 Foote et al., 1980; 41 Lechner et al., 1997]. The activation of LC in response to intestinal and pelvic stimuli is correlated with cortical electroencephalographic indices of excitement, such as desynchronization and a shift from high-rise, low-frequency activity at low intensity, high-frequency activity [42 Page et al., 1992]. The Barrington's nucleus neurons with their projections to the spinal cord and LC integrate then signals from multiple pelvic organs and coordinates their activity with the activity in prosencephalous regions underlying behavior. CRF is indeed expressed in Barrington's nucleus neurons. Specifically, CRF axons from Barrington's nucleus end both to LC and parasympathetic preganglionic nucleus, which innervates the pelvic region. Physiologically, Barrington’s nucleus neurons expressing CFR respond to both bladder and colon distension. Through the projections of the Barrington’s nucleus to the LC, CRF has mainly excitatory effects and modulates neuronal activation in LC in response to pelvic visceral stimuli which, in turn, arouse cortical excitement [43 Lechner et al., 1997]. On the other hand, through the spinal projections from the Barrington’s nucleus, CRF have excitatory effects on colon motoneurons, helping to increase the motility of colon [44 Bellman, 1966].

In general then, it is conceivable that alterations of the complex system involving the Barrington’s nucleus might be involved in specific pathophysiological mechanisms underlying elimination disorders such as enuresis and fecal incontinence. Considering the frequent association between fecal incontinence, enuresis and sleep disturbances, it is interesting to note that LC is associated with the hypocretin/orexin system, which plays indeed a key role in sleep/wake regulation. In support of the anatomo-functional connection between LC and hypocretin/orexin system it has been shown that administration of hypocretins directly in LC increases wakefulness and reduces REM sleep [45 Bourgin et al., 2000].

Conversely, in Barrington’s nucleus, Orexin receptors are expressed as described by immunohistochemistry analysis [46 Greco and Shiromani, 2001] supporting a possible relevant role of the orexins system in sleep alterations of FNFRI children as described in our study.

In the medical literature, there are currently very limited amounts of data reporting sleep quality in children with fecal incontinence. Considering the social impact of this disorder, a better understanding of the pathophysiological mechanisms underlying this disturb is crucial. In this regard, further investigations are needed to improve possible preventive tools and expand therapeutic strategies.

We have taken into account some limitations of the present study such as lack of follow-up evaluation, which could represents indeed a second step of the current study. However, we also would like to emphasize one of the points of strength of this report that consisted in the relatively large number of FNFRI children enrolled and that this is one of the first report describing alterations of sleep PSG and Orexin-A abnormalities in this type of pediatric patients. Also, we propose that the analysis of sleep habits, PSG recordings and Orexin-A plasma levels, may be considered as an affordable clinical tool for the screening and confirmation of FNFRI in children. However, further studies are necessary to establish specificity and sensitivity of the PSG recordings and their orexin-system plasma correlates in the context of a complex disorder of the development, which remains still to be better understood.

Reviewer 2

 Nonetheless, the manuscript required to improve by reviewing a few minor concerns.

Thank you for your comment. I modified all the text as your suggestion. 

Round 2

Reviewer 1 Report

The authors have made considerable changes to the manuscript. However, some comments have been left unanswered or aspects of the manuscript have been left unchanged despite my suggestions, without a clear argument why the authors disagree with the feedback provided. Therefore, I still think major revisions are needed. In the response by the authors, a lot of text has been copy-pasted from the manuscript, in the future, it would be more valuable for the authors to refer to the manuscript and to use the reply section for arguments and explanations. 

Although the language has improved, minor editing of English language and style is still required.

Although the terminology has been changed, this has not been carried out in the entire document. The 1st sentence of the introduction is an example ("soiling"). Please check the manuscript thoroughly. 

The authors have not used the correct reference for pediatric Rome IV criteria. Please use Hyams et al Gastroenterology. 2016 Feb 15. pii: S0016-5085(16)00181-5. (for older children). 

Introduction:

The authors state: "Enuresis and urinary incontinence are less common as comorbidity." In FNRFI, the opposite is true in my opinion and based on the literature I am aware of. Please provide a reference or change this statement. 

Methods:

I previously made this comment: "The authors have chosen to use a statistical analysis for normally distributed data. I strongly doubt the data were indeed normally distributed with such small sample sizes, if the data are not normally distributed a different test should have been used (Mann Whitney U). This is very important for the entire paper. Moreover, if data are not normally distributed, I suggest reporting the median, IQR and range rather than the mean and SD." In their reply, the authors have not addressed the content of my comment in any way. Please look into this aspect with care, if the applied statistical analyses are not appropriate for the dataset, the data may be unreliable.

I previously made this comment: "The tables and text are full of abbreviations, making it impossible to easily read through and comprehend the results without constantly looking at the meaning of the abbreviations. A further explanation of the results is required here." Not much has changed, I therefore encourage the authors to address this comment once more. 

Discussion:

I previously made this comment: "The discussion contains quite a lot of repetition. On the other hand, the authors hardly discuss the actual main findings of the study: the association between abnormal sleep and FNRFI. This deserves more attention." The authors have hardly changed anything to this section.

Author Response

Response to Reviewer

The authors have made considerable changes to the manuscript. However, some comments have been left unanswered or aspects of the manuscript have been left unchanged despite my suggestions, without a clear argument why the authors disagree with the feedback provided. Therefore, I still think major revisions are needed. In the response by the authors, a lot of text has been copy-pasted from the manuscript, in the future, it would be more valuable for the authors to refer to the manuscript and to use the reply section for arguments and explanations. 

A; We thank Reviewer for valued suggestions and for appreciation of our modified manuscript

Although the language has improved, minor editing of English language and style is still required.

A: The manuscript was checked and revised in each section carefully

Although the terminology has been changed, this has not been carried out in the entire document. The 1st sentence of the introduction is an example ("soiling"). Please check the manuscript thoroughly. 

A: We are sorry for mistakes. We have checked and modified as suggested the wrong terms.

The authors have not used the correct reference for pediatric Rome IV criteria. Please use Hyams et al Gastroenterology. 2016 Feb 15. pii: S0016-5085(16)00181-5. (for older children). 

A: We have added the correct reference n. 20 i(References section)

Introduction:

The authors state: "Enuresis and urinary incontinence are less common as comorbidity." In FNRFI, the opposite is true in my opinion and based on the literature I am aware of. Please provide a reference or change this statement. 

A; We erased the uncorrect period

Methods:

I previously made this comment: "The authors have chosen to use a statistical analysis for normally distributed data. I strongly doubt the data were indeed normally distributed with such small sample sizes, if the data are not normally distributed a different test should have been used (Mann Whitney U). This is very important for the entire paper. Moreover, if data are not normally distributed, I suggest reporting the median, IQR and range rather than the mean and SD." In their reply, the authors have not addressed the content of my comment in any way. Please look into this aspect with care, if the applied statistical analyses are not appropriate for the dataset, the data may be unreliable.

A: We thank Reviewer for the valued and precious suggestions about the correct statistical analysis to use. We recalculated and corrected the stats, the tables and the results sections

I previously made this comment: "The tables and text are full of abbreviations, making it impossible to easily read through and comprehend the results without constantly looking at the meaning of the abbreviations. A further explanation of the results is required here." Not much has changed, I therefore encourage the authors to address this comment once more. 

A: We have specified also in tables and in the text the abbrevations

Discussion:

I previously made this comment: "The discussion contains quite a lot of repetition. On the other hand, the authors hardly discuss the actual main findings of the study: the association between abnormal sleep and FNRFI. This deserves more attention." The authors have hardly changed anything to this section.

A: We thank Reviewer for suggestions. We have added the more relevant findings in the Discussion section and we have modified the section substantially.